# Transformable masks for colloidal nanosynthesis

Zhenxing Wang[1,2,3,4], Bowen He[1], Gefei Xu[3], Guojing Wang[3], Jiayi Wang[5], Yuhua Feng[3], Dongmeng Su[3], Bo Chen[6], Hai Li[7], Zhonghua Wu[5], Hua Zhang [6], Lu Shao[2] & Hongyu Chen[1,3]

Synthetic skills are the prerequisite and foundation for the modern chemical and pharmaceutical industry. The same is true for nanotechnology, whose development has been hindered by the sluggish advance of its synthetic toolbox, i.e., the emerging field of nanosynthesis. Unlike organic chemistry, where the variety of functional groups provides numerous handles for designing chemical selectivity, colloidal particles have only facets and ligands. Such handles are similar in reactivity to each other, limited in type, symmetrically positioned, and difficult to control. In this work, we demonstrate the use of polymer shells as adjustable masks for nanosynthesis, where the different modes of shell transformation allow unconventional designs beyond facet control. In contrast to ligands, which bind dynamically and individually, the polymer masks are firmly attached as sizeable patches but at the same time are easy to manipulate, allowing versatile and multi-step functionalization of colloidal particles at selective locations.

[1] Institute of Advanced Synthesis, School of Chemistry and Molecular Engineering, Jiangsu National Synergetic Innovation Center for Advanced Materials, Nanjing Tech University, Nanjing 211816, China. [2] MIIT Key Laboratory of Critical Materials Technology for New Energy Conversion and Storage, State Key Laboratory of Urban Water Resource and Environment, School of Chemistry and Chemical Engineering, Harbin Institute of Technology, Harbin 150001, China. [3] Division of Chemistry and Biological Chemistry, School of Physical and Mathematical Sciences, Nanyang Technological University, 21 Nanyang Link, Singapore 637371, Singapore. [4] College of Chemistry, Nanchang University, Nanchang 330031, China. [5] Institute of High Energy Physics, Chinese Academy of Sciences, University of Chinese Academy of Sciences, Beijing 100049, China. [6] Center for Programmable Materials, School of Materials Science and Engineering, Nanyang Technological University, Singapore 639798, Singapore. [7] Key Laboratory of Flexible Electronics and Institute of Advanced Materials, Jiangsu National Synergetic Innovation Center for Advanced Materials, Nanjing Tech University, Nanjing 211816, China. Zhenxing Wang and Bowen He contributed equally to this work. Correspondence and requests for materials should be addressed to L.S. (email: shaolu@hit.eud.cn) or to H.C. (email: iashychen@njtech.edu.cn)

The emerging field of nanosynthesis focuses on the development of synthetic skills at the nanoscale[1–6]. One major bottleneck at the moment is the design of chemical selectivity on nanoparticles. Unlike organic compounds with distinct functional groups, colloidal particles have only facets and ligands, which are highly inter-dependent, as many facets only exist in the presence of specific ligands. This complexity makes it extremely difficult to manipulate them as synthetic handles, more specifically for the following reasons. Firstly, there is no method available to control the area, shape, and position of ligand patches[7]. While facet-specific functionalization of Au nanorods (AuNRs) has been achieved[8–10], controlling ligand patches beyond facets remains a major challenge. Secondly, the ligands are never permanent—they dynamically bind and dissociate from the nanoparticle surface, even for the strongest thiol ligands on Au/Ag surfaces[11]. Hence, some degree of mixing is inevitable. Thirdly, the colloidal instability of nanoparticles is a constant nuisance, which not only narrows the choice of ligands but also poses massive uncertainties in any exploration attempts.

Decorating nanoparticles with single[12] and multiple island domains[1,13,14] have been achieved, but efforts of exploiting such islands as synthetic masks were still rudimentary[9,15,16]. While the structural simplicity is one reason, a more important reason is that the island masks are stationary and only variable from the synthesis. Hence, their regio-selective role can only be exploited once. Much can be desired if the mask is adjustable for multi-step regio-selective reactions.

Herein, we demonstrate that polymer shells on nanorods can undergo different modes of shell transformation, allowing then to act as versatile synthetic handles beyond facet control. The polymer shells are highly resistant to aggregation but at the same time fluidic and adjustable, an essential feature for achieving site-selectivity in multi-step nanosynthesis.

## Results

**Concept**. The critical issue is to find a stable but fluid mask that can be precisely manipulated, so that the encapsulated nanoparticles can be selectively exposed and functionalized. Here, different structures are derived from heat-induced transformation of polystyrene- block-poly(acrylic acid) (PSPAA) shells, which are created by a general method of nanoparticle encapsulation[17]. The PSPAA-coated nanoparticles are known to be extremely stable, even in saturated salt solutions[18]. To start, AuNRs were mixed with the amphiphilic PSPAA and a hydrophobic ligand (**1–6**, Fig. 1) in dimethyl formamide (DMF)/$H_2O$ mixture (4.5:1 V/V), and then heated at 110 °C for 2 h, giving core-shell structures ((AuNR-ligand)@PSPAA, Supplementary Figs. 1–5). Basically, the PSPAA self-assembles on the hydrophobically functionalized nanoparticle surface, with PS blocks facing inward and the PAA blocks dissolved in the solvent. These nanoparticles were purified repeatedly to remove DMF and the excess ligand, and then dispersed in water. To our great interest, simple heating can transform the resulting PSPAA shells in different modes (contraction, bimodal contraction, dissociation, and winding), depending on the choice of ligand and their concentration in the initial encapsulation step.

**Transformation modes**. For (AuNR-**1**)@PSPAA, heating in water caused contraction of the polymer domain, and its degree can be precisely controlled by adjusting the temperature and duration of this step. At 105 °C over 1.5–4 h, the polymer domain slowly contracts (Fig. 2b–d), gradually exposing the AuNRs from the ends and revealing increasing area of the side facets (6 ± 2%, 12 ± 4%, and 27 ± 3%, see Fig. 2r). The polymer appears to be stuck at this stage, as longer heating (105 °C for 6 h) does not cause further exposure. Only with harsher conditions, we

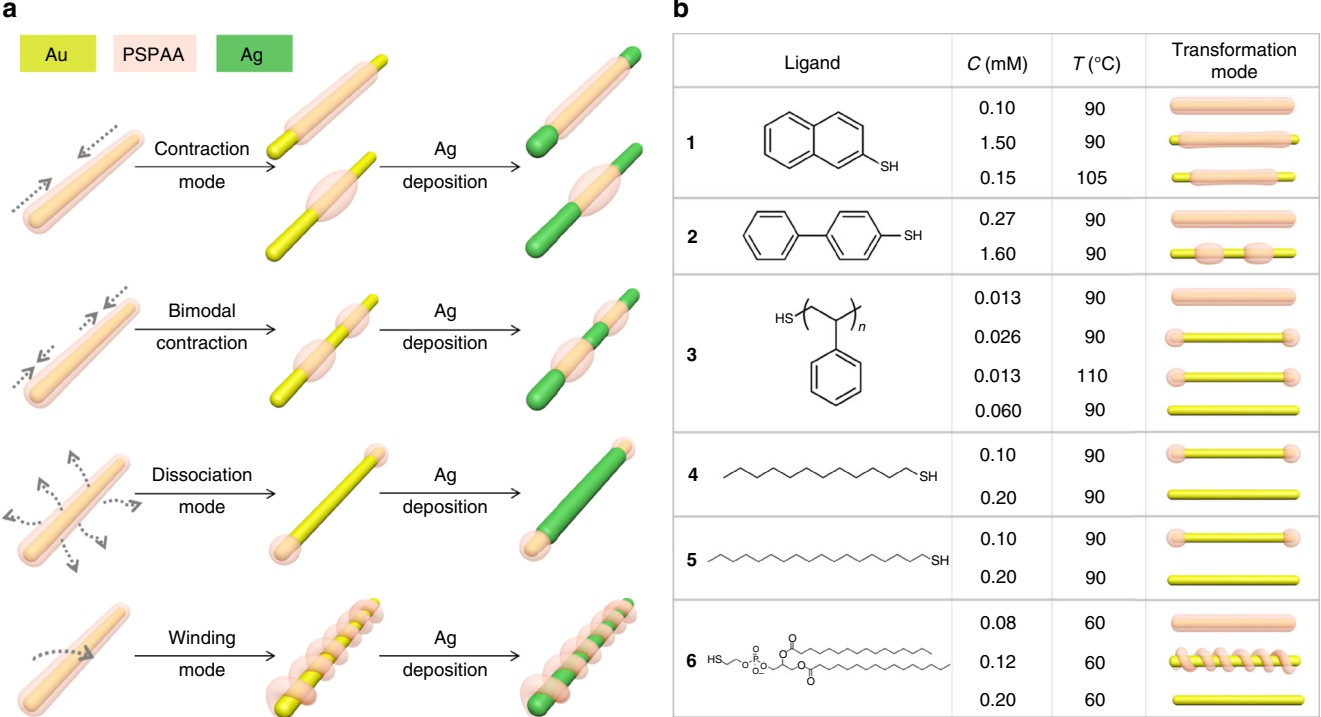

**Fig. 1** Schematics illustrating the four transformation modes of (AuNR-ligand)@PSPAA. **a** Contraction, bimodal contraction, dissociation, and winding modes (left panel). **b** The ligand dependence is summarized in the Table on the right panel. Ligands **1–6** are 2-naphthalenethiol, biphenyl-4-thiol, thiol-terminated polystyrene, 1-dodecanethiol, 1-octadecanethiol, and a thiol-ended phospholipid, 2-dipalmitoyl-*sn*-glycero-3-phosphothioethanol (sodium salt), respectively. C (mM) the concentration of the ligand in the initial encapsulation step, T the temperature used for transforming the (AuNR-ligand)@PSPAA in water

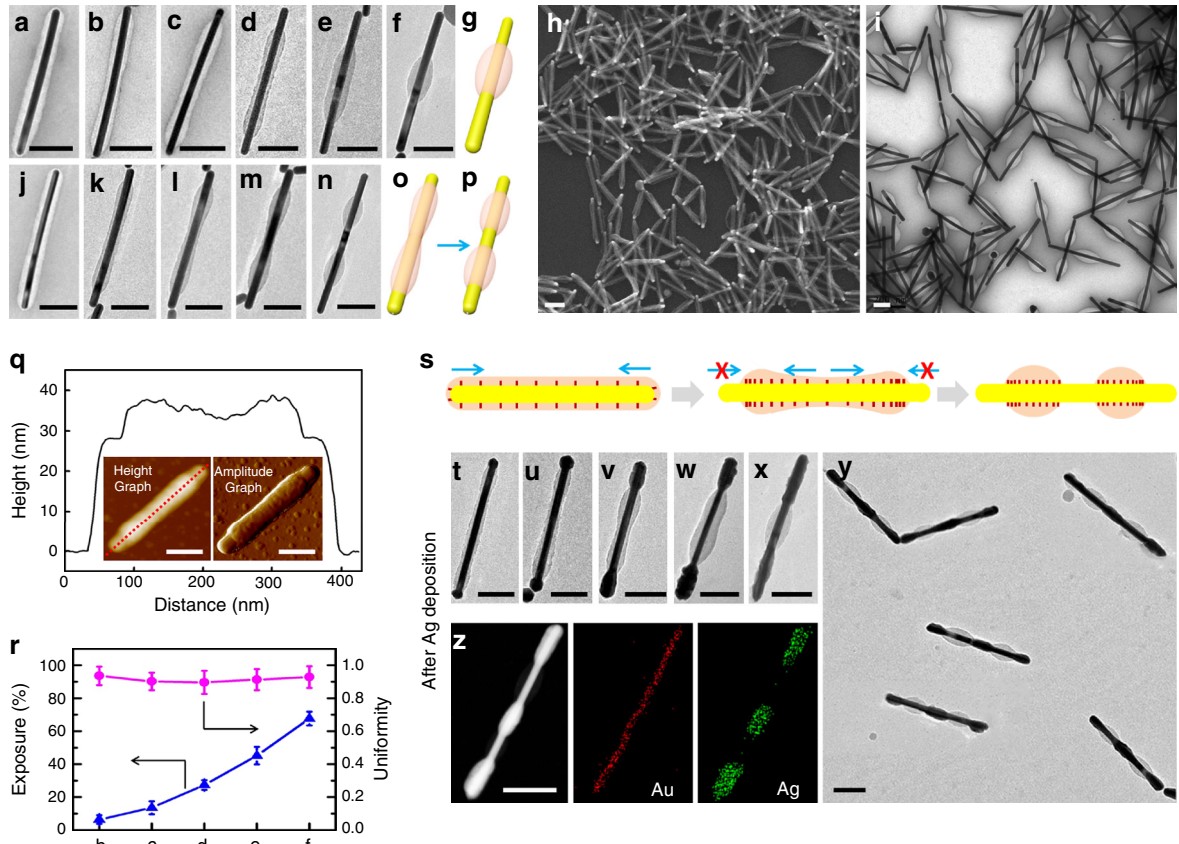

**Fig. 2** Contraction and bimodal contraction modes. TEM images of **a** the initial core-shell (AuNR-**1**)@PSPAA; **b**–**d** the (AuNR-**1**)@PSPAA after 1.5 h **b**, 2.5 h **c** and 4 h **d** heating at 105 °C in water; **e** after 3 h heating in water at 110 °C; **f** and **i** after 3 h heating in water at 115 °C, with their structure illustrated by **g**. **h** SEM image of the sample **c**. **j**–**n** (AuNR-**2**)@PSPAA before **j** and after bimodal contraction at 90 °C for 3 h **k**, and the selected intermediates (**l**, **m**) towards the product at 6 h **n**. **o**, **p** Schematics illustrating the shell transformation from **m** to **n**. **q** AFM image and line scan of the (AuNR-**1**)@PSPAA in the sample **h**. **r** the average degree of exposure and the ratio of the two exposed ends (uniformity = $l_{short}/l_{long}$) in the samples of **b**–**f** (based on at least 100 AuNRs). Error bars, s.d. (n = 3). **s** Schematics illustrating the possible mechanism of bimodal contraction, where the crowded ligand restricts further contraction from the ends. **t**–**y** TEM images of the above masked nanostructures after Ag deposition on the exposed Au surface, with **t**–**x** corresponding to **b**–**f** and **y** corerponding to **n**. **z** STEM image and EDX mapping for Au and Ag on the (AuNR-**2**)@PSPAA after Ag deposition, respectively. The scale bars are 100 nm. Large-area images are shown in the Supplementary Figs. 9–12, 17,18, 24–31

managed to expose 45 ± 5% (110 °C for 3 h, Fig. 2e) and 68 ± 5% of the side facets (115 °C for 3 h, Fig. 2f, g). Large-view scanning electron microscope (SEM) image of the sample 1c (Fig. 2h) and TEM image of the sample 1 f (Fig. 2i) demonstrate the purity of the typical samples, and the structure was further supported by atomic force microscope (AFM) line scan (Fig. 2q). It appears that the high temperatures also caused significant dissociation of the shells, leading to reduced volume of the polymer domain. On the other hand, at lower temperature of 90 °C, the contraction was just enough to expose the tips of the AuNRs, but the progress was barely noticeable even after 12 h (Supplementary Fig. 6).

The partial polymer shells on these exposed AuNRs are highly uniform in terms of the extent of contraction and the equal distance from the two ends (Fig. 2h, i, r and Supplementary Figs. 7–12). Figure 2b appears similar to the facet-specific functionalization in the literature[8,9], but the subsequent structures offer a unique ability to gradually expose the side facets of the AuNRs (Fig. 2b–f), going beyond facet specificity.

The contraction of the PSPAA shell is found to be shape-dependent. For Au nanospheres (AuNSs), even the harshest condition (115 °C for 3 h) did not cause exposure of the Au surface in (AuNS-**1**)@PSPAA (Supplementary Fig. 13). We speculate that the PSPAA shell above its glass transition temperature (100 °C for bulk PS)[19] may behave like a viscous

liquid as the gradual de-wetting exposes the high curvature ends. More specifically, withdrawing the spindle-like shells of (AuNR-**1**)@PSPAA is favorable in terms of reducing the PS-water interface, but the shell on (AuNS-**1**)@PSPAA is already spherical. Breaking it would initially cause a dramatic increase of the interface. Indeed, the surface tension of the PSPAA shell on AuNRs is evident even in the core-shell stage before the contraction (Fig. 2a), where the shell at the tips is much thinner than that covering the sides.

Interestingly, ligand **2** led to a bimodal contraction mode. When the (AuNR-**2**)@PSPAA (Fig. 2j) were heated in water at 90 °C, the polymer shell first contracted to expose the two tips (3 h, Fig. 2k). Then, the two polymer ends got stuck after this stage and the middle section of the polymer shell became stretched and thinner (Fig. 2l). These behaviors were similar to the above case. However, with longer heating, a break-up point is reached (Fig. 2m, o) and the polymer shell split into two segments, which contracted further from the middle (6 h, Fig. 2n, p). The bimodal contraction is uniform and reproducible, and it depends on the initial ligand concentration (1.6 mM) at the encapsulation step. When less ligand (0.27 mM) was used, the polymer shell showed minimal contraction even after 6 h at 90 °C (Supplementary Fig. 14). We estimated that completely covering all Au surface in our experiments would require only 0.2 µM ligand and hence, the

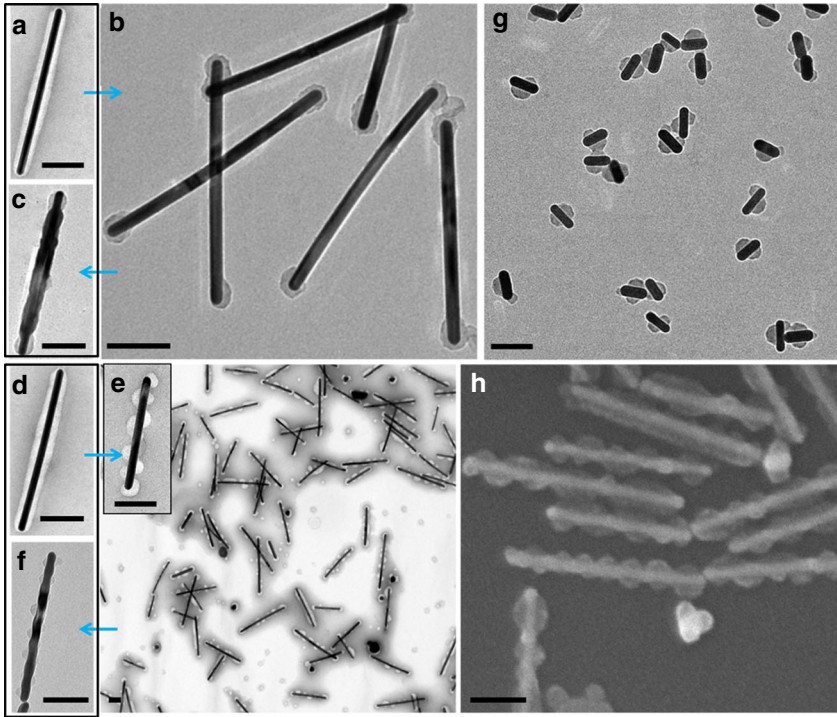

**Fig. 3** Dissociation and winding modes. **a**, **b** TEM images of the (AuNR-**3**)@PSPAA before **a** and after **b** heating in water at 90 °C for 3 h. **c** TEM image of the transformed (AuNR-**3**)@PSPAA after Ag deposition on the exposed Au surface. **d**, **e** TEM images of the (AuNR-**6**)@PSPAA before **d** and after **e** heating in water at 60 °C for 3 h. The inset in **e** is the TEM image of the transformed (AuNR-**6**)@PSPAA with high magnification. **f** TEM image of the transformed (AuNR-**6**)@PSPAA after Ag deposition. **g** The shorter (AuNR-**6**)@PSPAA after shell transformation. **h** SEM image of the transformed (AuNR-**6**)@PSPAA. The scale bars are 100 nm

ligands used were in large excess. Considering that free ligands in solution would have been completely removed during purification, they would have no effect in the shell contraction step. The only way is via the excess ligands remained in the shell[20], likely as a plasticizer[21] improving the liquidity of the polymer (vide infra).

The high percentage of the products with bimodal contraction is intriguing. The different parts of the polymer shell are expected to have similar liquidity under the same reaction conditions. However, the two ends of the polymer section after the initial contraction (exposing the tips) cannot move further, whereas the middle section still can. On this basis, we hypothesize that the ligand **2** may migrate on the Au surface along with the contracting polymer shell and became denser, preventing further movement of the shell (Fig. 2s). This hypothesis is consistent with (1) the expectation that the hydrophobic ligand should be retained within the hydrophobic polymer domain; (2) the colloidal stability of the contracted particles in water, suggesting the hydrophilic exposed surface; (3) the fact that Ag and Pd can be readily deposited on the exposed Au surface without ligand exchange (vide infra).

When ligand **3** was used, a dissociation mode was observed: most of the polymer shell on (AuNR-**3**)@PSPAA disappeared after heating in water at 90 °C for 3 h, leaving behind only two caps on the tips of the AuNRs (Fig. 3a, b). The extent of polymer dissociation depends on the ligand concentration (0.026 mM) during the initial encapsulation step. With higher ligand concentration (0.06 mM), the polymer shell completely disappeared after 3 h at 90 °C (Supplementary Fig. 15). At a lower concentration (0.013 mM), 90 °C heating had little effects (Supplementary Fig. 16), and harsher conditions (105 °C for 3 h) were necessary to achieve the structure of Fig. 3b (Supplementary Fig. 17).

Most interestingly, the use of amphiphilic ligand **6** led to helical polymer domains after the shell transformation (the winding mode, Fig. 3d, e, h, and Supplementary Fig. 18). For the (AuNR-**6**)@PSPAA, the shell dissociation was easier than the above cases and hence the heating was carried out at only 60 °C for 3 h. The concentration of ligand **6** during the encapsulation step was critical: At 0.12 mM, the polymer domain partially dissociated and the remaining polymer became a helical cylinder winding around the AuNRs (Supplementary Fig. 18). At higher ligand concentration (>0.2 mM), most of the polymer shell disappeared after the heating (Supplementary Fig. 19); at lower concentration (0.08 mM), neither dissociation nor transformation was observed (Supplementary Fig. 20). This winding mode can be also achieved on short AuNRs ($l = 76 \pm 3$ nm, Fig. 3g and Supplementary Fig. 21), where the remaining polymer domain appeared dissymmetrical after heating in water, suggesting similar winding mode. For ligands **1**–**3**, the winding mode cannot be achieved despite repeated trials with varying ligand concentration and heating temperature.

**Determining factors**. The reaction conditions leading to the four transformation modes are summarized in Fig. 1 (see greater details in Supplementary Fig. 22). Several general trends can be recognized: (1) when the ligand concentration or temperature is below threshold, the polymer transformation cannot occur; (2) excess ligand and high temperature are complimentary in promoting transformation; and 3) when allowed, the transformation is more extensive with longer incubation time.

Among the ligands, **1** appears to be the strongest. The contraction of the polymer shell with ligand **1** is independent of the initial ligand concentration and higher temperatures are necessary. We speculate that the strong π–π interaction among the aromatic ligands might be responsible. Ligand **2** would be similar to ligand **1** in terms of close packing, except that the twisted conformation of the biphenyl rings would partially

compromise the stability of the resulting patches. Indeed, bimodal contraction can also be achieved with (AuNR-**1**)@PSPAA, except that harsher conditions (120 °C for 6 h) are necessary and the yield was much lower (<10%) as compared to (AuNR-**2**) @PSPAA. In comparison, the polymeric ligand **3** would be too bulky to pack well on the Au surface, facilitating the polymer shell dissociation. The polymer cap at the tips of AuNR has a high contact area with Au per unit volume of polymer, allowing it to persist longer than the shell on the flat facets. To further investigate the effects of ligand packing, we tried to use **4** and **5** as ligand. At 0.10 mM, the dissociation mode was observed (90 °C for 3 h, Supplementary Fig. 23), and at 0.20 mM all polymer shell was completely lost (Supplementary Fig. 24). It appears that these ligands without π–π interaction dissociate more readily from the Au surface[22].

Considering the lowest temperature of transformation, ligand **6** appears to be the weakest. It is important to note its amphiphilicity—having ionic and polar functionalities underneath the hydrophobic PS layer may cause the ligand to be less stable, or more mobile. Importantly, with **6** at the Au-PS interface and the tethered PAA blocks at the PS-water interface, the polymer domain is enclosed by surfactants. As such, it is a micelle, except that it adsorbs on the Au surface with the thiol ends, as opposed to the conventional micelles freely suspended in solution. Transforming the polymer shell into a winding cylinder is not favorable in terms of reducing the PS solvent or Au-PS interface. It appears that the polymer domain prefers a cylindrical shape driven by the micelle properties[5,19], and the resulting cylinder would be longer than the length of the AuNR, forcing it to wind around. The chiral mechanism is similar to the insertion of a wire into a cylinder, where the confined environment forces the initial tilting and a helical shape ensues as a result of energy minimization, because forming sharp kinks is more costly than smooth winding[23]. Similarly, the polymer cylinder confined on the surface of the AuNR has to randomly tilt at one direction, and the remaining cylinder would wind around following the same direction.

On these bases, we summarize the determining factors in Fig. 4. The polymer liquidity appears to be a prerequisite, without it the ligands cannot exert influence as they are locked at their original place by the glassy polymer. It is known that low-molecular-weight molecules such as solvents and ligands can

dissolve in the PS domain, where the former is known as the swelling agent and the latter as plasticizer[21]. These molecules form secondary bonds (van der Waals interactions, and so on) with the PS chains and spread them apart. Thus, they weaken the PS–PS interactions and provide more mobility for the PS domain, resulting in a softer mass with a lower glass transition temperature. Hence, both high temperature and high ligand concentration can improve polymer liquidity and they are complimentary in promoting shell transformation. Only with sufficient polymer liquidity, the incubation time would have effects and the same is true for the ligands. Ligands with strong π–π interaction are more difficult to dissociate and probably also interact with PS chains better, allowing them to persist on the Au surface and leading to the contraction modes. Aliphatic and bulky ligands would dissociate upon heating, causing the entire polymer patches to dislodge and move into the aqueous phase as free micelles. On the other hand, amphiphilic ligand can cause the polymer domain to be micelle-like, leading to winding cylinders. In short, ligand is the critical factor determining the modes of shell transformation, whereas excess ligand, high temperature, and time affect polymer liquidity.

**Masks for colloidal nanosynthesis.** The partially exposed AuNRs provide a versatile platform for further functionalization, where the distinct Au and PSPAA surfaces can be exploited for chemical selectivity. As a demonstration, we deposited Ag on the above masked AuNRs. In a typical synthesis, the masked AuNRs were purified from the preparative solution, without further ligand exchange. They were mixed with the reductant hydroquinone, followed by AgNO$_3$, so that the resulting Ag atoms can be deposited on the AuNRs. For all of the four masked structures discussed above, Ag was selectively deposited on the exposed Au surface, judged by the obvious thickening at the corresponding segments (Fig. 2t–x, Fig. 3c, f, and Supplementary Figs. 25–32). There was no Ag deposited on the PSPAA surface. Figure 2y shows the AuNRs with bimodal contraction after growing Ag, giving Ag-Au-Ag-Au-Ag penta-block structure. Energy-dispersive X-ray spectroscopy (EDX) mapping in Fig. 2z confirms the selective deposition of Ag.

The most important aspect of continual contraction is to exploit the different stages for chemical selectivity of multi-step reactions. As a proof-of-concept, we coated Pd on the exposed Au surface and then induced further contraction of the polymer. As illustrated in Fig. 5a, the second step contraction revealed a small gap of Au surface between the Pd-coated and the PSPAA-coated domains, where Ag can be selectively deposited. Specifically, the reduction of Na$_2$PdCl$_4$ by hydroquinone led to a conformal coating on the exposed Au surface[24] and a layer of fine particles on the PSPAA shell (Fig. 5b–e). The latter made the PSPAA shell appear dirty and gave rise to a smeared Pd region in the EDX mapping (Fig. 5d). After the second contraction, we performed Ag deposition by reacting AgNO$_3$ with hydroquinone, and obtained a structure that resembles a blot with two nuts (Fig. 5f). The selective thickening at the neck position of all of the AuNRs implies the success of our synthetic strategy, likely because Ag has matching lattice with Au, but is mismatched with Pd[25]. The Ag domain was confirmed by EDX mapping as shown in Fig. 5e. When AuNRs with longer exposed segments (similar to Fig. 2d) were used in step 1, the thickened neck position appeared closer to the center (Fig. 5g) as expected. To the best of our knowledge, selective deposition at arbitrary location on AuNRs has not been achieved to date. The highly predictable nature of the PSPAA shell transformation allows rational design of multi-step, regio-selective chemical reactions.

PAA can be embedded within inorganic domains (Au, PbTiO$_3$, Fe$_3$O$_4$, CdSe, and so on) so that PAA-based molecules can serve

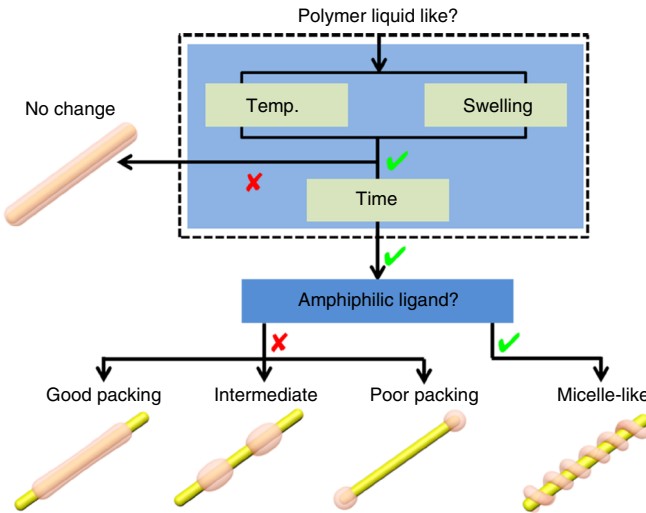

**Fig. 4** Flow chart summarizing the determining factors. Ligand is the critical factor determining the modes of shell transformation, whereas excess ligand, high temperature, and time affect polymer liquidity

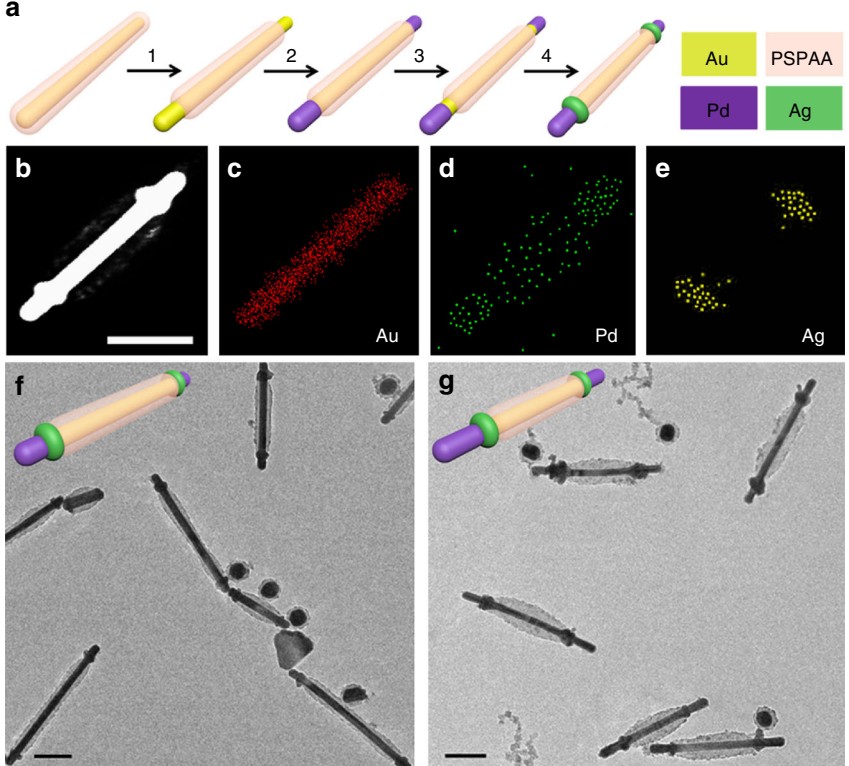

**Fig. 5** Multi-step masked nanosynthesis. **a** Schematics illustrating the multi-step functionalization enabled by the continual contraction of the PSPAA shell. **b**–**g** STEM image, EDX mapping, and TEM image of the resulting $Pd_{tip}$-$Ag_{ring}$-(AuNR-**1**)@PSPAA-$Ag_{ring}$-$Pd_{tip}$, with short **b**–**f** and long **g** exposed segments. The scale bars are 100 nm

as template for the growth of inorganic nanoparticles[26]. This capability can be attributed to the interaction of PAA with metal ions through the −COOH groups. In our system, we choose conditions so as to avoid metal deposition (Ag and Pd) on the PAA surface. Since the metal–metal bonds are in general stronger than the −COO-metal interaction, it would be possible to suppress the metal nucleation on the −COOH surface by competition. In other words, metal can be selectively deposited on the exposed metal surface, rather than the PAA surface. Between Ag and Pd, Ag atoms are better stabilized in solution than Pd atoms, and Pd has larger lattice mismatch with Au (meaning less effective Pd–Au bonds at the interface). Hence, Pd deposition is in general less selective than Ag deposition in our system.

Welding colloidal structures together is a major challenge in the field, as the crosslinking reagents would typically compromise the colloidal stability of nanoparticles, but the surfactants introduced to solve the problem would compromise the effects of crosslinking. Considering the dilemma, the partially masked AuNRs may lead to a breakthrough, as the partial PSPAA shell can prevent aggregation and the exposed tips can lead to selective points of contact. Specifically, we coated the (AuNR-**1**)@PSPAA with Ag tips (similar to Fig. 2u). The product was dispersed in water, followed by the crosslinking reagent $Na_2S$, and the mixture was kept at room temperature for 2 h. The $Na_2S$ caused the nanorods to aggregate via the exposed tips, and then converted the Ag layer to $Ag_2S$, crosslinking the nanorods in the process (Fig. 6a, Supplementary Figs. 33 and 34), which can be further supported by the small angle X-ray scattering (SAXS) results (Supplementary Fig. 35 and Supplementary Table 1). The inset in the Fig. 6b shows that the two exposed ends were embedded in a single $Ag_2S$ nanocrystal. STEM image and corresponding EDX mapping confirmed the composition assignments (Fig. 6c, d).

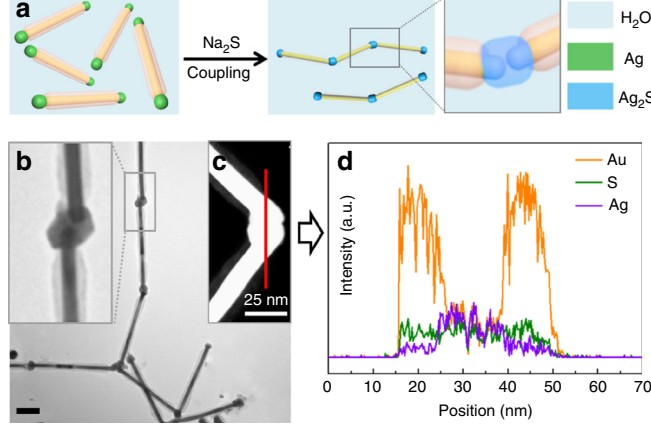

**Fig. 6** Welding of AuNRs via $Ag_2S$ bridges. **a** Schematics illustrating the end-to-end welding of (AuNR-**1**)@PSPAA in colloidal solution. **b** TEM image of the welded AuNRs. **c**, **d** STEM image and EDX line scan for the welded junction. The scale bar is 100 nm

Thus, with PSPAA shells at the specific locations, the tip-to-tip aggregation and the welding of the nanorods via $Ag_2S$ bridges can be readily carried out. In comparison, doing the same for hexadecyltrimethylammonium bromide (CTAB)-stabilized AuNRs encountered significant hurdles of reagent compatibility because of the labile surfactant.

We applied the polymer masks on Au nanoparticles with different shapes. As shown in Fig. 7, the tips of Au bipyramid (Fig. 7a, and Supplementary Figs. 36–38) and triangular nanoprisms (Fig. 7b, and Supplementary Fig. 40) can be selectively exposed by contracting the PSPAA shells. With the help of polymer masks, Ag can be selectively deposited on the

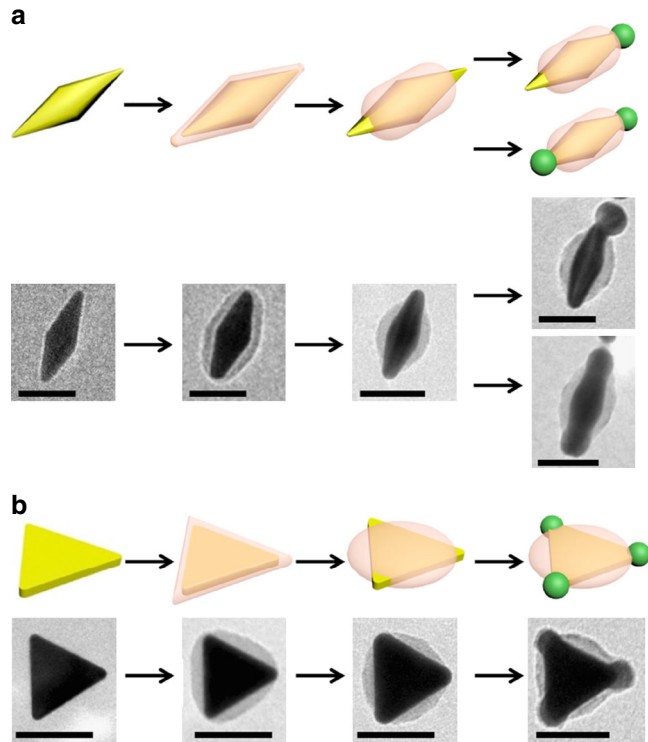

**Fig. 7** Bipyramids and triangular nanoprisms. **a**, **b** Schematics and TEM images illustrating the encapsulation and contraction of PSPAA on Au bipyramids **a** and Au nanoprisms **b**, and the corresponding Ag deposition, respectively. The ligand used in this experiment is ligand-**1**. The scale bars are 50 nm

exposed Au surface, giving two and three tips, respectively. Interestingly, gold bipyramid with single Ag tip (Supplementary Fig. 38) can be obtained when the rate of Ag reduction was slow; and growth on both tips can be achieved (Supplementary Fig. 39) when the reduction was fast. This asymmetric Ag overgrowth behavior can be explained by our previously reported depletion sphere mechanism[14,27], where the slower build-up of Ag oversaturation can lead to a larger radius of depleted sphere of Ag atoms, inhibiting growth at the other tip.

## Discussion

In conclusion, we exploit the multiple modes of PSPAA shell transformation for masked synthesis of colloidal nanorods. The transformation modes are governed by the type of ligand and its concentration in the encapsulation step, provided that the polymer shell has sufficient liquidity. Unlike solid inorganic shell, the pliable PSPAA shell can move and transform, so that its rich interaction with the underlying ligand layer can be fully exploited, leading to diverse structures. Moreover, the gradual receding of the polymer shell gives multiple layers of reaction sites—an unprecedented synthetic capability with similar roles as the protecting groups in organic chemistry. The polymer surface with long, ionic PAA chains renders the nanoparticles with great colloidal stability and a certain degree of inertness during metal deposition. The polymer masks are firmly attached to the nanoparticles, so that there are few compatibility issues with the subsequent reagents used for metal deposition and crosslinking.

With such a synthetic platform, new colloidal syntheses can be designed, where the stable polymer patches and their predictable behavior are critical in realizing regio-selectivity and multi-step reactions. Our six-step colloidal synthesis, including the synthesis of nanorods, encapsulation, contraction, Pd deposition, further

contraction, and Ag deposition, is a powerful, though preliminary, demonstration of colloidal synthetic designs. We believe that further exploitation of the pliable polymer masks can bring about a leap forward in sophisticated nanosynthesis.

## Methods

**Materials and methods**. All chemical reagents were used without further purification. Hydrogen tetrachloroaurate (III) hydrate 99.9% (metal basis Au 49%), 1-octadecanethiol (96%) and 1-dodecanethiol (98%) were purchased from Alfa Aesar; sodium citrate tribasic dihydrate (99.0%), hydroquinone (99.5%), AgNO$_3$ (99.9999%), 2-naphthalenethiol (99%), biphenyl-4-thiol (97%), hexadecyl-trimethylammonium bromide (99%), and cetyltrimethylammoniumchloride (CTAC, 99%) were purchased from Sigma-Aldrich; 1, 2 dipalmitoyl-*sn*-glycero-3-phosphothioethanol (sodium salt) was purchased from Avanti Polar Lipids. Amphiphilic diblock copolymers polystyrene-*block*-poly (acrylic acid) (PS$_{154}$-*b*-PAA$_{49}$, $M_n$ = 16,000 for the PS block and $M_n$ = 3500 for the PAA block, $M_w/M_n$ = 1.15) and thiol-terminated polystyrene (PS-SH, $M_n$ = 2000, $M_w/M_n$ = 1.15) were purchased from Polymer Source Inc.; *N,N* dimethyl formamide (DMF, 99.8%) was purchased from Merck; 200 mesh copper specimen grids with formvar/carbon support film (referred to as TEM grids in text) were purchased from Electron Microscopy Sciences. Deionized water (DI water, resistance > 18.2 MΩ per cm) was used for all experiments.

TEM images were collected from a JEM-1400 (JEOL) Transmission Electron Microscopy operated at 100 kV. High-resolution TEM (HRTEM) image was taken from JEOL 2100F field-emission transmission electron microscope at 200 kV. SEM image was collected from a field-emission scanning electron microscopy (FE-SEM, JEOL JSM-7001F). AFM (Dimension ICON, Bruker) was employed to observe the nanorods in tapping mode in air. X-ray scattering experiments were performed at SWAXS Xenocs Nanoinxider with $\lambda$ = 0.154 nm. The sample-to-detector distance for SAXS was 938 mm. Fit2D software was used to analyze the obtained SAXS data.

**Preparation of (AuNR-ligand)@PSPAA (aspect ratio is about 18.0 ± 1.2)**. AuNRs with high aspect ratio were prepared according to seed-mediated synthesis by Murphy and co-workers[28]. The AuNRs obtained from 50 mL of solution were collected and purified by centrifugation and washed by DI water for four times to remove the excess CTAB (Supplementary Fig. 41).Then the purified AuNRs were dispersed in 2.5 mL of DI water to obtain the AuNR solution (stored at 5 °C before use). The encapsulation method was modified from our previously published method with minor modifications. In a typical encapsulation process, 130 µL of AuNR solution and 70 µL of DI water were added into a mixture prepared by mixing DMF (820 µL) and PSPAA (80 µL, 8 mg/mL in DMF). Then thiol ligand such as 2-naphthalenethiol (**1**) or biphenyl-4-thiol (**2**) or thiol-terminated polystyrene (**3**) or 1-dodecanethiol (**4**), or 1-octadecanethiol (**5**), or 1, 2 dipalmitoyl-sn-glycero-3-phosphothioethanol (sodium salt) (**6**) was added in the above mixture. Finally, the mixture was heated at 110 °C for 2 h, and then allowed to slowly cool down till room temperature. The concentration of the ligand in the encapsulation step was shown in Supplementary Fig. 22.

**Preparation of (AuNR-6)@PSPAA (aspect ratio is about 2.8 ± 0.5)**. Au nanorods with low aspect ratio were prepared using method by Murphy and co-workers[29]. For encapsulation, 1 mL of obtained pristine AuNR solution was centrifuged at 9800×*g* for 10 min to obtain ~10 µL concentrated solution. The concentrated solution was then diluted by 1 mL of DI water, and centrifuged at 9800×*g* for 8 min to obtain ~10 µL concentrated solution. This process was repeated twice to remove the excess CTAB. The washed AuNR was diluted by DI water to 200 µL, and then added into a mixture that was prepared by mixing DMF (820 µL) and PSPAA (80 µL, 8 mg/mL in DMF). Then 60 µL of ligand-**6** (2 mg/mL in EtOH) was added into the above mixture. Finally, the mixture was heated at 110 °C for 2 h, and then allowed to slowly cool down till room temperature.

**Preparation of (AuNS-1)@PSPAA**. Gold nanospheres (AuNS, about 40 nm in diameter) were prepared according to literature procedures by sodium citrate reduction of HAuCl$_4$[30]. Citrate stabilized AuNS solution was centrifuged at 9800×*g* for 10 min to obtain ~10 µL concentrated solution. The concentrated solution was then diluted by DI water (200 µL), and then added into a mixture that was prepared by mixing DMF (820 µL) and PSPAA (80 µL, 8 mg/mL in DMF). Then ligand-**1** (5 µL, 10 mg/mL in DMF) was added in the above mixture. Finally, the mixture was heated at 110 °C for 2 h, and then allowed to slowly cool down till room temperature.

**Preparation of (bipyramid-1)@PSPAA**. Gold bipyramids were prepared according to the literature[31]. One milliliter of pristine bipyramid solution was centrifuged at 5200×*g* for 8 min to obtain ~10 µL concentrated solution. The concentrated solution was then diluted by 1 mL of CTAB (1 mM), and centrifuged at 4000×*g* for 6 min to obtain ~10 µL concentrated solution (this process was repeated twice to remove the HCl). The 10 µL of concentrated solution was diluted to 0.5 mL with DI water, and was centrifuged at 2000×*g* for 8 min to obtain ~10 µL concentrated solution to remove the excess CTAB. The washed bipyramid was diluted by DI

water to 200 µL, and then added into a mixture that was prepared by mixing DMF (820 µL) and PSPAA (80 µL, 8 mg/mL in DMF). Then 5 µL of ligand-**1** (10 mg/mL in DMF) was added into the above mixture. Finally, the mixture was heated at 110 °C for 2 h, and then allowed to slowly cool down till room temperature.

**Preparation of (nanoprisms-1)@PSPAA**. Triangular gold nanoprisms were prepared using method by Zhang and co-workers[32]. One milliliter of pristine nanoprism solution was centrifuged at 5200×$g$ for 6 min to obtain ~10 µL concentrated solution. The concentrated solution was then diluted by 1 mL of CTAB (1 mM), and centrifuged at 5200×$g$ for 5 min to obtain ~10 µL concentrated solution (this process was repeated twice to remove the CTAC). The 10 µL of concentrated solution was diluted to 0.8 mL with DI water, and centrifuged at 2000×$g$ for 5 min to obtain ~10 µL concentrated solution. The washed nanoprisms were diluted by DI water to 200 µL, and then added into a mixture that was prepared by mixing DMF (820 µL) and PSPAA (80 µL, 8 mg/mL in DMF). Then 5 µL of ligand-**1** (10 mg/mL in DMF) was added into the above mixture. Finally, the mixture was heated at 110 °C for 2 h, and then allowed to slowly cool down till room temperature.

**Transformation of (Au-ligand)@PSPAA**. The obtained (Au-ligand)@PSPAA mixture (1 mL) was centrifuged for 8 min to a volume of ~10 µL (4000×$g$ for (AuNR-ligand)@PSPAA and 9800×$g$ for (AuNS-**1**)@PSPAA)). The concentrated mixture was diluted by 600 µL of DI water and then centrifuged to a volume of ~10 µL (this process was repeated twice to remove the residual DMF and ligand). Finally, to induce the transformation of PSPAA, the concentrated mixture was diluted by 0.6 mL of DI water and heated at designed temperature for different duration (Supplementary Fig. 1, Supplementary Fig. 22). The transformed (Au-ligand)@PSPAA can be directly used for the next metal deposition.

Supplementary Fig. 42 shows the experimental setup in an airing chamber for the transformation of (AuNR-ligand)@PSPAA, and the table shows the inside/inner pressure of the bottle at different heating temperature. The cubage of the little bottle is about 4 mL, and there is a sealing pad in the bottle cap to avoid the gas leakage. The bottle can bear the pressure even if the water was heated to 120 °C. But for security sake, the glass screen of the airing chamber should be pulled down during the experiment.

**Silver deposition on transformed (AuNR-ligand)@PSPAA**. In a typical synthesis, 5 µL of hydroquinone (1 mM) was added into 0.6 mL of transformed (AuNR-ligand)@PSPAA suspension. After vortex for several seconds, 5 µL of AgNO$_3$ (1 mM) was added into the above mixture. The mixture was left undisturbed for 12 h at room temperature. Specially, for constructing two silver tips on the transformed (bipyramid-1)@PSPAA, the pH value of the mixture was adjusted from 7 to 11, without changing the concentration of hydroquinone and AgNO$_3$.

**Preparation of (AuNR-1)@PSPAA-Pd tips-Ag rings**. In a typical synthesis, 0.6 mL of water dispersed (AuNR-**1**)@PSPAA was heated at 105 °C for 2 h and cooled down to room temperature. Then 5 µL of hydroquinone (10 mM) and 5 µL of Na$_2$PdCl$_4$ (10 mM) were added. The mixture was undisturbed at room temperature for 12 h to prepare Pd-tipped (AuNR-**1**)@PSPAA. After being centrifuged to a volume of ~10 µL, 0.6 mL of water was added into the concentrated Pd-tipped AuNR@PSPAA suspension, and heated at 110 °C for 3 h to further contract. After cooled down to room temperature, 5 µL of hydroquinone (1 mM) and 5 µL of AgNO$_3$ (1 mM) was added, and stand for 12 h to form silver rings on the new exposed gold surface. The length of Pd tips and the position of Ag ring can be controlled by increasing the first heating duration to 3 h.

**Coupling of (AuNR-1)@PSPAA-Ag**. Firstly, (AuNR-**1**)@PSPAA-Ag was prepared based on the contracted (AuNR-**1**)@PSPAA (the contraction duration is about 2 h). The obtained solution containing (AuNR-**1**)@PSPAA-Ag was centrifuged at 2000×$g$ for 5 min to a volume of ~10 µL. Then 0.5 mL of DI water was added followed by adding 5 µL of sodium sulfide (1 mM). The mixture was kept for 2 h at room temperature to realize the coupling of AuNRs.

**Preparation of TEM samples**. TEM grids were pretreated for 30 s in a Harrick plasma cleaner/sterilizer to improve their hydrophilicity. The hydrophilic face of the TEM grid was then placed in contact with the sample solution. Filter paper was used to wick off the excess solution on the TEM grid, which was then left to dry in air for 10 min. For some samples, (NH$_4$)$_6$Mo$_7$O$_{24}$ was used as the negative stain, so that the polymer shells appear white against the stained background.

**Preparation of SEM samples**. A silicon wafer (about 0.6 cm$^2$) was pretreated with O$_2$ plasma for 15 min to improve its surface hydrophilicity. The wafer was then functionalized with an amino group by reacting with APTES solution (5 mM) for 30 min. Subsequently, the wafer was washed by EtOH for several times. Water dispersed sample was dropped on the wafer and stand for 30 min. Then the redundant solution on the wafer was drawn off, and the wafer with sample was kept at room temperature until dry.

**Data availability**. All data generated or analyzed during this study are present in the main text and the Supplementary Information. Additional data are available from the authors on reasonable request.

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

## Acknowledgements

We thank the MOE (RG 14/13) of Singapore, the National Natural Science Foundation of China (nos. 21673117, 21676063, and U1462103), Jiangsu Provincial Foundation for specially-appointed professor, start-up fund at Nanjing Tech University, and HIT Environment and Ecology Innovation Special Funds for financial supports. Z.W. thanks the support from China Scholarship Council (CSC) program. G.W. acknowledges the Tsinghua Scholarship for Overseas Graduate Studies. B.H. acknowledges the Jiangsu National Synergetic Innovation Center for Advanced Materials (SICAM) Scholarship for Overseas Graduate Students.

## Author contributions

H.C., L.S. and Z.W. designed and planned this project, and proposed the mechanisms. Z. W. was responsible for the preparation of contraction and winding modes. G.X. was responsible for the dissociation mode. B.H. purified the AuNRs to high purity, and helped in the syntheses and characterization of (AuNR-**1**)@PSPAA and (AuNR-**6**) @PSPAA. G.W. and Y.F. helped in the syntheses of metal tipped colloidal nano-composites. D.S. helped in the SEM characterization. H.L. helped in the AFM characterization. J.W. and Z.W. helped in the SAXS characterization. B.C. and H.Z. helped in the HRTEM characterization. All authors participated in the writing of the manuscript.

## Additional information

**Competing interests:** The authors declare no competing financial interests.

