## [Peer Review File · Nature Communications]

Reviewers' comments:

Reviewer #1 (Remarks to the Author):

In this manuscript, Wang et al. demonstrated an interesting approach to the synthesis of colloidal Au nanorods regioselectively decorated with another metal using phase-separated polymer shell as a transformable mask. This synthesis appears to be a versatile method to produce a range of multicomponent nanorods (e.g., Ag/Au or Ag/Pd/Au nanorods) with different compositional arrangements. This paper is well written and the nanostructures are well characterized. However, some major concerns prevent me to be more enthusiastic about the work. Therefore, I would recommend the publication of this manuscript in Nature Communications, only after the following comments are addressed:

1. A huge library of colloidal particles (or rods) has been produced in the past decade and the current work is the icing on the cake. However, it is not clear to me what are the unique properties or applications of these nanorods, particularly when the size of these nanorods is big. The authors showed the fusion of Ag tipped Au nanorods, but this phenomenon is well known and again it is not clear what properties the fusion of rods will lead to. It is critical to look into the properties and to demonstrate at least one killing application of these rods. For example, will the chiral coating of Ag on rods exhibit any unique optical property?
2. PAA is known to be able to interact with metal ions and has been widely used as templates for metal deposition (see Nature Nanotechnology, 2013, 8, 426-431, etc.). It is unclear why the deposition of metals did not happen on PAA phases in this work. A more convincing description of the mechanism should be given.
3. It is unclear how generic this method will be. Both Ag and Pd were used as the secondary metal. However, as shown in the EDX mapping of the Pd/Ag/Au nanorods (Figure 3), the deposition of Pd occurred everywhere on the surface of rods including the surface of polymer domains. It is crucial to demonstrate that this method is applicable for producing other types of multicomponent nanorods with good regioselectivity.
4. Figure 3, Pd deposited on the surface of both polymer and Au domains. What causes the difference in the selectivity of Ag and Pd deposition? This should be stated and discussed in the main text.
5. What is the limit in the domain size of polymer masks? In another word, is this approach applicable to other shaped nanoparticles (e.g., cubes) or smaller nanoparticles (e.g., 40 nm nanospheres)? Is it possible to produce Ag/Au nanorods with various morphologies using shorter nanorods?

Reviewer #2 (Remarks to the Author):

This paper describes a very creative approach to the creating of (mostly) gold nanorods coated with patches of different molecules/metal. The great thing about the paper is the creativity shown in reaching the goal. I would also like to stress that the results shown are really nice and captivating. There are two issues with the paper. The first is somewhat minor, there has been a long standing interest in assembly of patchy particles (probably 15 years now). While progress in the assembly has been made there are few truly convincing applications for these assemblies. Of course, a lot of applications are not possible if the assembly are not solid, scalable, and reproducible. Based on these considerations, I would give this paper a pass, even though it does not contain any hint of a convincing applications for this paper.

The true major point for this paper is in the characterisation approach taken to prove its point. I counted a total of less than 200 rods shown in the sum total of the TEM images that support the paper conclusions. No image contains more than 10 rods. Honestly this is way too little to convincingly prove the paper's point.

If the assembly is a key point of the paper than it need to be supported by SAXS, DLS and other bulk

methods. As for the selective dewetting of the polymer I would suggest the authors to try creating monolayers of the rods and using AFM to show their point over a large number of rods. If the chemistry proposed works on a small subtraction of the rods, then the paper is not publishable in Nature Communications.

Point-by-Point Reply

Manuscript ID: NCOMMS-16-30088A

Title: Transformable Masks for Colloidal Nanosynthesis

Reviewer #1:

Comments: In this manuscript, Wang et al. demonstrated an interesting approach to the synthesis of colloidal Au nanorods regioselectively decorated with another metal using phase-separated polymer shell as a transformable mask. This synthesis appears to be a versatile method to produce a range of multicomponent nanorods (e.g., Ag/Au or Ag/Pd/Au nanorods) with different compositional arrangements. This paper is well written and the nanostructures are well characterized. However, some major concerns prevent me to be more enthusiastic about the work. Therefore, I would recommend the publication of this manuscript in Nature Communications, only after the following comments are addressed:

1. A huge library of colloidal particles (or rods) has been produced in the past decade and the current work is the icing on the cake. However, it is not clear to me what are the unique properties or applications of these nanorods, particularly when the size of these nanorods is big. The authors showed the fusion of Ag tipped Au nanorods, but this phenomenon is well known and again it is not clear what properties the fusion of rods will lead to. It is critical to look into the properties and to demonstrate at least one killing application of these rods. For example, will the chiral coating of Ag on rods exhibit any unique optical property?

Response: We thank the reviewer for the recognition and insightful questions. We believe that this work is a major advance in nanosynthesis, rather than just an additional new structure to the current collection (“icing on the cake”).

I would argue that *breakthroughs in synthesis are as important as those in application*. Modern pharmaceutical industry and all of its benefits should owe their existence 99% to the foundation of chemistry knowledge and 1% (or even less) to the “crown jewel” of drug discoveries. The same would be true for nanotechnology. Unfortunately, the current field is too occupied by application that few researchers devote their career to making synthetic advances.

The problem is not that aiming for the “crown jewel” is wrong, but that a field with everyone trying to build the top floor and no one building the basement is unhealthy. *There should be the “crown jewel” of application as well as the “crown jewel” in building the foundation.*

I agree with the reviewer that “a huge library of colloidal particles (or rods) has been produced” and that they are often too similar to be of importance. It is exactly because of this that we (researchers, funding agencies, and journals) should promote *new types of advances* in nanosynthesis.

Without question the current nanotechnology is far from the promised Future Nanotechnology in fictions and our imagination. We would only fool ourselves if we believe that continuing the current path of applying nanowires and nanorods, without making synthetic advances, would lead to a Future full of nanorobots and sophisticated nanodevices. The gap is wide and obvious.

It would be a daunting task to unify people's opinions on what constitutes a major synthetic advance, particularly in a new field of nanosynthesis and when everyone is waiting for *new structures* that can make killer applications. But in my humble opinion, *new synthetic capabilities* are more crucial to the field of synthesis, as the combination of capabilities would lead to *rational synthesis* and easily give rise to millions of structural variations.

It is on the solid foundation of synthetic capabilities (as opposed to the mere collection of molecules) that modern pharmaceutical industry thrives; and it is not surprising that many researchers in nanotechnology fall short of their dream breakthroughs as they are limited by the current synthetic capability. The dreams would remain as dreams as long as we cannot even make simple "nuts and bolts" at the nanoscale and put them together.

The narrow scope of the existing nanosynthesis is almost laughable and yet few people seem to worry about it. To make advances, I believe that we should develop *new capabilities* of making multi-component nanostructures, of joining particles together, of *regioselective growth/attachment (the focus of this work)*, of reducing symmetry in nanostructures, of creating new types lattices, so on and so forth. We are proud that our works in the past 10 years focused on these *new capabilities* rather than simply expanding the collection of *new nanostructures*.

Our current work on the transformable masks in colloidal synthesis is a major advance in synthesis, because *regioselective growth* has been a bottleneck in creating sophisticated nanostructures. Without this capability, construction of complex nanostructures can only be done in the dark. The strategy of relying on *facet specificity* only gave *symmetrical additions* to a particle and often led to messy results due to ligand incompatibility (see

detailed discussions in the manuscript). With this capability, we can start to design new complex structures and achieve it with multi-step *rational synthesis*. Indeed, our *6-step colloidal synthesis*, including the synthesis of nanorods, encapsulation, contraction, Pd deposition, further contraction, and Ag deposition, is a powerful, though preliminary, demonstration what *rational design can* do to *change colloidal synthesis*.

We understand that our view of the field is unusual and perhaps ahead of the time. We are confident that the above arguments are logical and wholistic, and would invite the reviewer to see the significance from our perspective. We hope that the reviewer would agree with us and appreciate our passion and efforts.

Comments: 2. PAA is known to be able to interact with metal ions and has been widely used as templates for metal deposition (see Nature Nanotechnology, 2013, 8, 426-431, etc.). It is unclear why the deposition of metals did not happen on PAA phases in this work. A more convincing description of the mechanism should be given.

Comments: 4. Figure 3, Pd deposited on the surface of both polymer and Au domains. What causes the difference in the selectivity of Ag and Pd deposition? This should be stated and discussed in the main text.

Response: We thank the reviewer for the insightful question. The trick is to control the oversaturation of the metal atoms during its deposition and consequently, the heterogeneous nucleation event. The critical point of nucleation varies on different surfaces, depending on the surface energy. In other words, it depends on the stability of atomic metal clusters on (*i.e.*, number and strength of bonds with) the surface. See our mechanistic review: *Angew. Chem.Int. Ed.*, **2015**, 54, 2022-2051.

We agree with the reviewer that PAA can interact with metal ions through –COOH groups. But since metal-metal bonds are in general stronger than the –COO-metal interaction, it would be possible to suppress the nucleation on the –COOH surface by competition. In our experiments, Ag atoms are better stabilized in solution than Pd atoms. In addition, Pd has larger lattice mismatch with Au, meaning less effective Pd-Au bonds at the interface. Hence, Ag deposition is in general more selective than Pd deposition in our system. **We have modified the text to reflect this point.**

In contrast, embedding PAA inside inorganic materials involves different challenges and mechanism, in terms of how large amount of organic defects are tolerated by the growing inorganic domains. I would expect the inorganics to deposit on inorganic surface around the PAA, rather than on top of the PAA.

Comments: 3. It is unclear how generic this method will be. Both Ag and Pd were used as the secondary metal. However, as shown in the EDX mapping of the Pd/Ag/Au nanorods (Figure 3), the deposition of Pd occurred everywhere on the surface of rods including the surface of polymer domains. It is crucial to demonstrate that this method is applicable for producing other types of multicomponent nanorods with good regioselectivity.

Comments: 5. What is the limit in the domain size of polymer masks? In another word, is this approach applicable to other shaped nanoparticles (e.g., cubes) or smaller nanoparticles (e.g., 40 nm nanospheres)? Is it possible to produce Ag/Au nanorods with various morphologies using shorter nanorods?

Response: We thank the reviewer for the great suggestions. When we initially planned this project, there were several directions we could possibly explore: 1) variety on the shape of polymer shells; 2) variety on the size/shape of the seeds; 3) variety on the composition of the seeds; 4) variety of the metals being deposited; 5) understanding the critical role of ligands; and 6) exploiting the contraction modes for nanosynthesis. Given the limited time of a project and limited scope/space of a manuscript, we have to choose among these directions. The shape of polymer shells became the theme of this work, because it can best demonstrate the ability to *control regioselective growth*. In addition, the role of the ligands provides critical supporting arguments and the use in

nanosynthesis provides the proof-of-concept. Considering the initial report of the colloidal masking approach, we are worried that discussion on diverse topics may dilute the key point.

We have now provided new data to demonstrate the generality of our strategy (**Fig. 5** and Supplementary **Fig. S32-36**). Besides AuNRs, we further investigate the application of polymer masks on other gold materials with different morphologies (**Fig. 5**). For example, we encapsulated gold bipyramids by PSPAA with 2-naphthalenethiol as ligand (Supplementary **Fig. S32**). The encapsulated core-shell structures were heated in water at 110°C for 2 h to induce the contraction mode, which exposed the two tips of bipyramid (Supplementary **Fig. S33**). With the help of polymer masks, silver can be selectively deposited on the exposed gold surface (**Fig. 5a**). Interestingly, gold bipyramid with single silver tip can be obtained when the concentrations of AgNO₃ and hydroquinone are both about 10 μM (Supplementary **Fig. S34**). This asymmetric Ag overgrowth behavior can be explained by our recently reported depletion sphere mechanism.^[1, 2] According to this mechanism, increasing the reduction rate would drive up the oversaturation at the critical point of nucleation, leading to more (two) silver tips on the bipyramid. Similarly, the reduction rate was increased by adjusting the pH of mixture from 7 to 11 (without changing the concentration of AgNO₃ and hydroquinone). Almost all of the bipyramids were decorated with two silver tips (**Fig. S35**). The polymer masks can be also applied to gold nanoplate to introduce three exposed tips (**Fig. 5b**, and Supplementary **Fig. S36**).

References:

- [1] Feng, Y., Wang, Y., Song, X., Xing, S. & Chen, H. Depletion sphere: Explaining the number of Ag islands on Au nanoparticles. *Chem. Sci.*, **2017**, 8, 430-436
- [2] Feng, Y. et al. Achieving Site-Specificity in Multistep Colloidal Synthesis. *J. Am. Chem. Soc.*, **2015**, 137, 7624-7627

Figure R1. Applicability of the pliable masks. Schematics and TEM images illustrating the encapsulation and contraction of PSPAA on Au bipyramids (a) and triangular gold nanoprisms (b), and the corresponding Ag deposition, respectively. The ligand used in this experiment is Ligand-1. The scale bars are 50 nm. (These images have been included in our revised manuscript, please see Fig. 5). The large-view images have been included in the Supporting Information.

We have also attempted polymer contraction on Ag, Pd, TiO₂ and Te nanowires, as shown below. Their different surfaces require distinct processing conditions and hence, it would be better to expand this topic in a separate paper.

Figure R2. TEM image of the pristine core-shell structure and transformed structures of (TiO₂-1)@PSPAA, (TeNR-1)@PSPAA, and (Ag-1)@PSPAA. The transformation was carried out in water at 115 °C for 2 h.

We are also exploring and expanding the method to deposit different metals, but for the purpose of creating more complex nanostructures. Instead of the 6-step treatment of PSPAA coated AuNRs, if we could achieve 20- or 30-step colloidal synthesis, I believe it would be another breakthrough in synthetic capability.

Reviewer #2:

Comments: This paper describes a very creative approach to the creating of (mostly) gold nanorods coated with patches of different molecules/metal. The great thing about the paper is the creativity shown in reaching the goal. I would also like to stress that the results shown are really nice and captivating.

There are two issues with the paper. The first is somewhat minor, there has been a long standing interest in assembly of patchy particles (probably 15 years now). While progress in the assembly has been made, there are few truly convincing applications for these assemblies. Of course, a lot of applications are not possible if the assemblies are not solid, scalable, and reproducible. Based on these considerations, I would give this paper a pass, even though it does not contain any hint of a convincing application for this paper.

Response: We thank the reviewer for the high recognition and the understanding that not all breakthroughs of synthesis can be directly exploited. As discussed in our above reply to Reviewer #1, we hope that the community would value synthetic advances as much as applications. With more synthetic capabilities accumulated, we (the field of nanotechnology) would then be able to design and synthesize for the purpose of application. *It is unfortunate that most of the fundamental knowledge in nanoscience was accumulated only as a by-product of application-oriented research (not oriented directly to expanding synthetic capabilities).*

Comments: The true major point for this paper is in the characterisation approach taken to prove its point. I counted a total of less than 200 rods shown in the sum total of the TEM images that support the paper conclusions. No image contains more than 10 rods. Honestly this is way too little to convincingly prove the paper's point.

Comments: If the assembly is a key point of the paper then it needs to be supported by SAXS, DLS and other bulk methods. As for the selective dewetting of the polymer I would suggest the authors to try creating monolayers of the rods and using AFM to show their point over a large number of rods. If the chemistry proposed works on a small subtraction of the rods, then the paper is not publishable in Nature Communications.

Response: We thank the reviewer for the critical point. The synthesis of Au nanorods was not pure following the literature method and our initial purpose was only the proof-of-concept demonstration of the synthetic capabilities. We agree with the reviewer that larger scale characterization is necessary to prove our point. Hence, we have purified the Au nanorods to high purity and repeated the masking methods. The TEM images are shown below. Obtaining such images with 50-100 rods would be impossible if the sample is not pure. Suppose that a sample is only 50% pure, meaning that each randomly sampled rod has 1/2 chance of being the intended one. Searching and cropping for a nice picture with 50-100 rods would easily take longer than one's lifetime. 2^{100} seconds is a trillion times the age of the universe.

We note that the bulk characterization methods for nanoparticles are much limited. DLS cannot distinguish the scattering contribution from the polymer shells in contrast to the strong scattering from the Au nanorods. AFM is similar and inferior to TEM in terms of large-area sampling. SAXS is a possible method but we have no access to synchrotron facility and it would be impossible to secure a timing slot with the short notice. Indeed, I would argue that advancing characterization methods is also as important as application, but unfortunately few researchers devote their career to it.

Figure R3. TEM images of the AuNRs before (a) and after (b) purification. (The images have been added into our Supporting Information, please see Supplementary Fig. 37)

Figure R4. TEM images of the (AuNR-1)@PSPAA after heating at 105°C in water for 2.5 h. (The images have been added into our Supporting Information, please see Supplementary Fig. 8)

Figure R5. TEM images of the (AuNR-1)@PSPAA after heating at 115 °C in water for 3 h. (These pictures have been added into our revised manuscript, please see Fig. 1 and Supplementary Fig. 11)

Figure R6. TEM images of the (short AuNR-1)@PSPAA after heating at 105 °C in water for 3 h.

Figure R7. TEM images of the (AuNR-6)@PSPAA after heating at 60 °C in water for 3 h. (The images have been added into our revised manuscript, please see Fig. 2 and Supplementary Fig. 17)

Reviewers' comments:

Reviewer #1 (Remarks to the Author):

I'm convinced by the authors' response to the comments raised by the referees and would recommend the publication of the revised manuscript as is in Nature Communications.

Reviewer #2 (Remarks to the Author):

The authors have not replied to my comments. The paper remains without any viable application in mind (and I remain that this is only a minor weakness of the paper), and has no supporting evidence to the TEM images (that I learn now -I might have missed it before- where of non purified syntheses before).

The argument of no access to synchrotron facilities is totally faulty anybody can send applications to all facilities in the world and have access to them.

As it stands this paper does not meet the standards of Nature Communications.

Point-by-Point Reply

Reviewer #1:

I'm convinced by the authors' response to the comments raised by the referees and would recommend the publication of the revised manuscript as is in Nature Communications.

Response: We thank the reviewer for the positive comments.

Reviewer #2:

Comments:

The authors have not replied to my comments. The paper remains without any viable application in mind (and I remain that this is only a minor weakness of the paper), and has no supporting evidence to the TEM images (that I learn now-I might have missed it before-where of non purified syntheses before). The argument of no access to synchrotron facilities is totally faulty anybody can send applications to all facilities in the word and have access to them. As it stands this paper does not meet the standards of Nature Communications.

Response: We thank the reviewer again for the understanding that not all breakthroughs of synthesis can be directly exploited. SEM, AFM and SAXS analyses have been carried out and the supporting evidences added to our revised manuscript.

We have now improved our method of sample preparation, so that large-view images were obtained for both SEM and AFM. The sample purity/homogeneity can be supported by the large number of uniform particles shown (> 100). Suppose that one tries to obtain a “nice” image from a sample that is actually 50% pure (1/2 probability in sampling one particle), getting an image with 100 “right” particle would give a probability of $(1/2)^{100}$, which is essentially impossible. 2^{100} second would be a trillion times the age of our universe.

In the SEM data, the polymer shells can be clearly visualized, showing the exposed tips. In the AFM data, it was difficult to view the polymer shells in zoom-out views. In the particles that we examined with zoom-in views, the polymer shells and the exposed tips can be observed, as shown in Figure R2. In Figure R3, a large number of tip-to-tip aggregation can be recognized, showing successful coupling with Ag₂S formation.

Figure R1. SEM images of the contracted (AuNR-1)@PSPAA. The scale bar is 100 nm. (These results have been integrated into Figure 1 and Supplementary Figure 9 in our revised manuscript)

Figure R2. AFM images of the contracted AuNR@PSPAA. a, large-scale AFM images (height graph and amplitude graph) and b, AFM images (height graph and amplitude graph) with high magnification of the contracted (AuNR-1)@PSPAA. c, A line scan from tapping mode AFM corresponding to the red line in b. (These results have been integrated into Figure 1 in our revised manuscript)

Figure R3. Typical TEM images (low magnification) of the tip-to-tip coupling of AuNRs with Ag₂S as the junction. (These results have been added into our revised manuscript as Supplementary Figure 33)

The tip-to-tip aggregation and the “welding” of the AuNRs are shown in Figure R3 and further supported by the following SAXS results.

Small angle X-ray scattering results

Small angle X-ray scattering (SAXS) intensities were collected with an X-ray wavelength of 0.15412 nm in Xenoc-Nanoindexer as shown in Figure R4. From the SAXS data, a SAXS-intensity vibration can be clearly observed and the vibration amplitude is damped with the increase of the q-vector. It appears that the scatterers in the samples are in a monodisperse system. Therefore, a cylinder model of the scatterers with narrow Gaussian distribution of the diameter suggested by the TEM images was used to simulate the SAXS data. The simulated SAXS curves and the fitting parameters are shown in Figure R4 and Table R1, respectively. For the as-synthesized AuNRs, the average diameter and length obtained by SAXS analysis (16 and 287 nm, respectively) match well with the size distribution obtained from TEM and SEM images (Figure R5, 17.6 and 298 nm, respectively). Comparison of the SAXS results before and after sulfur-induced coupling showed that the average diameter of cylindrical particles remained unchanged at 16 nm, whereas their average length increases from 297 to 488 nm after the coupling. The Gaussian distribution widths (σ) of the particles slightly increase from about 0.9 nm to 1.2 nm after coupling. Hence, the data

agrees with the TEM observations in Figure R3, where the coupling of the AuNRs leads to longer average length and wider length distribution.

Figure R4. Experimental (circles) and fitting (solid lines) SAXS curves before and after coupling. (These results have been added into our revised manuscript as Supplementary Figure 34)

Table R1. SAXS fitting parameters with a cylinder model. D and L are the diameter and the length of the cylinder, respectively.

Samples	D (nm)	σ (nm)	L (nm)
Before coupling	16.0	0.9	297.0
After coupling	16.0	1.2	488.0

(These results have been added into our revised manuscript as Supplementary Table 1)

Figure R5. The statistical data (based on 247 rods) of the diameter and length of the pristine AuNRs

Reviewers' comments:

Reviewer #2 (Remarks to the Author):

The author have met my requirements. Now the paper is taking shape. The authors before publishing have to improve the fitting of the SAXS curve after assembly. The curve for $q > 1\text{nm}^{-1}$ has a slope that is not at all captured by the fit at the moment. Without a correct interpretation of this, their paper could be flawed.

Point-by-Point Reply

Manuscript ID: NCOMMS-16-30088B

Title: Transformable Masks for Colloidal Nanosynthesis

Reviewer #2:

Comments:

The authors have met my requirements. Now the paper is taking shape. The authors before publishing have to improve the fitting of the SAXS curve after assembly. The curve for $q > 1 \text{ nm}^{-1}$ has a slope that is not at all captured by the fit at the moment. Without a correct interpretation of this, their paper could be flawed.

Response: Thanks very much for your comments and suggestion. We have improved the fitting of the SAXS curve, and explained why the pervious curve ($q > 1 \text{ nm}^{-1}$) has a slope that is not captured by the fit at the moment. The revised fitting and explains are shown as follows:

Figure R1. Experimental (circles) and fitting (solid lines) SAXS curves (a) without and (b) with Porod correction before and after coupling. (The Figure R1b has been added into our revised manuscript, please see Supplementary Figure 34)

From **Figure R1(a)** (our previous fitting), it can be seen that two experimental data for $q > 1 \text{ nm}^{-1}$ have some positive divergence from fitting curves, which implies that the scattering intensities deviate positively from the Porod law because of the thermal diffuse scattering from the tiny

electron density fluctuation of particles. After multiplying the scattering intensities by $\exp(-Bq^2)$, $q^4I(q)$ were corrected to a constant. Then the Porod-corrected scattering data can be fitted very well as shown in **Figure R1(b)**. The Porod correction parameter B is also listed in Table R1.

Table R1. SAXS fitting parameters with a cylinder model. D and L are the diameter and the length of the cylinder, respectively. σ is the Gaussian distribution widths of the cylinder-particle radii. B is the porod correction parameter.

Samples	D (nm)	σ (nm)	L (nm)	B (nm ²)
Before	16.0	0.9	297.0	0.2
After	16.0	1.2	488.0	0.3

These results have been added into our revised manuscript as Supplementary Table 1

REVIEWERS' COMMENTS:

Reviewer #2 (Remarks to the Author):

I am not an expert on SAXS to the point where I can judge whether the Porod correction applied is the correct approach for sure, but it seems a good response to my concern. At this point I think the paper should be published.